# Masticatory dysfunction in patients with diabetic neuropathy: A cross-sectional study

**Yuta Hamamoto**[ID][1]*, **Kazuhisa Ouhara**[1], **Tsuyoshi Miyagawa**[2], **Tomoaki Shintani**[3], **Nao Komatsu**[1], **Mikihito Kajiya**[1], **Shinji Matsuda**[1], **Tsuyoshi Fujita**[1], **Shinya Sasaki**[1], **Tomoyuki Iwata**[1], **Haruya Ohno**[ID][4], **Masayasu Yoneda**[4], **Noriyoshi Mizuno**[1], **Hidemi Kurihara**[1]

**1** Department of Periodontal Medicine, Graduate School of Biomedical and Health Sciences, Hiroshima University, Hiroshima, Japan, **2** Clinical Research Center in Hiroshima, Hiroshima University Hospital, Hiroshima, Japan, **3** Central Clinical Divisions, Center of Oral Clinical Examination, Hiroshima University Hospital, Hiroshima, Japan, **4** Department of Molecular and Internal Medicine, Graduate School of Biomedical and Health Sciences, Hiroshima University, Hiroshima, Japan

* hamamoto@hiroshima-u.ac.jp

**Data Availability Statement:** All relevant data are within the paper and its Supporting Information files.

## Abstract

### Introduction

Chewing well is essential for successful diet therapy and control of blood glucose level in patients with diabetes. In addition, long-term hyperglycemia is a risk factor for microvascular complications, which are the main cause of morbidity and mortality in these patients. Hence, it is plausible that masticatory disorder may be relevant to diabetic microvascular complications which is caused by long-term hyperglycemia. The aim of this study was to investigate whether masticatory disorders are relevant to diabetic microvascular complications.

### Methods

This cross-sectional study included 172 patients with type 2 diabetes who underwent educational hospitalization in the Department of Endocrinology and Diabetic Medicine, Hiroshima University Hospital, from April 2016 to March 2020. Masticatory efficiency was determined quantitatively by using the GLUCO SENSOR GS-II. Multivariable linear regression models were constructed to examine which factors were related to masticatory efficiency. Statistical significance was defined as a two-sided $p$ value of $< 0.05$.

### Results

According to the bivariable analysis, masticatory efficiency was significantly correlated with duration of diabetes ($p = 0.049$), number of remaining teeth ($p < 0.0001$), the number of moving teeth ($p = 0.007$) and condition of diabetic neuropathy ($p < 0.0001$). Moreover, the number of remaining teeth ($p < 0.0001$) and diabetic neuropathy ($p = 0.007$) remained significantly correlated with masticatory efficiency in the multivariable analysis.

**Funding:** YH was supported by JSPS KAKENHI Grant Numbers JP19K24073 and JP20K18537. KO was supported by JSPS KAKENHI Grant Numbers JP18K09599. TM was supported by JSPS KAKENHI Grant Numbers JP20K18830. The funders played no role in the study design, data collection and analysis, decision to publish, or preparation of the manuscript.

**Competing interests:** The authors have declared that no competing interests exist.

## Conclusions

For the first time, we demonstrated that patients with type 2 diabetes who developed diabetic neuropathy had significantly reduced masticatory efficiency. Effective mastication is an important factor in successful diet therapy for diabetes. To prevent the progression of diabetic complications, especially in patients with diabetic neuropathy, it may be necessary to combine individualized therapies from dentists and nutritionists with consideration for the level of masticatory dysfunction.

## 1. Introduction

The oral cavity is the first line of the gastrointestinal tract and a portal of entry for microbes and foods. Since the oral cavity is constantly exposed to a plethora of external factors (e.g., microorganisms, mechanical and chemical damage from food intake, and dietary and airborne antigens), its function is frequently disrupted. Dysfunction of the oral cavity by bacterial infection, inflammation, tooth loss or pain has been suggested to negatively affect general health [1, 2]. Periodontitis is a representative disorder of the oral cavity that is characterized by a tissue-destructive chronic inflammatory disease caused by periodontal pathogenic microorganism infection. Previous studies have clearly demonstrated the correlation between periodontitis and diabetes; inflammation developed due to periodontitis has been associated with insulin resistance and poor glycemic control [3–6]. In addition, improvement of local inflammation after periodontal treatment positively affected insulin resistance and hemoglobin A1c (HbA1c) in diabetes patients [7–9].

Periodontitis is also characterized by alveolar bone destruction and subsequent tooth mobility or loss, leading to masticatory dysfunction [10–12]. It is widely accepted that tooth loss is associated with diabetes [13, 14]. Substantial mastication induces histamine production in the hypothalamus and insulin secretion from the pancreas, thereby preventing overeating and increasing blood glucose levels [15–18]. In previous studies, patients who lost their teeth suffered from malnutrition and preferred soft meals mainly composed of fats and carbohydrates [19–25]. An epidemiological study indicated that chewing slowly and steadily reduced the risk of diabetes [26]. Taken together, these results suggest that chewing sufficiently is essential for the success of diet therapy for diabetes and that long-term masticatory disorders is associated with the progression of diabetic conditions. Nonetheless, there are few studies investigating masticatory efficiency quantitatively in diabetes patients.

On the other hand, long-term hyperglycemia is well known as the major risk factor for vascular complications, which are the major cause of morbidity and mortality in diabetic patients [27]. Both microvascular and macrovascular complications affect the quantity and quality of life of diabetes patients. In particular, microvascular complications are responsible for neuropathy, retinopathy and nephropathy in diabetic patients. In addition, glycemic control delays the onset and progression of microvascular complications [28, 29]. Here, as described above, it is plausible that masticatory disorders may be associated with the progression of diabetic conditions. Moreover, it is true that uncontrolled diabetes causes subsequent microvascular complications. Thus, we speculate that masticatory disorder may be relevant to diabetic microvascular complications, although no study has verified this relationship.

In general, many previous reports have assessed masticatory efficiency by occlusal bite force, masticatory movement, or the ability to comminute test food [30–32]. Especially, assessing the mandibular kinematic of the chewing patterns is useful for understanding the

neuromuscular function [33]. However, it is difficult to assess the masticatory performance quantitatively by these previous methods. Hence, we focused on a new testing system that can quantitatively determine masticatory efficiency by measuring the eluted glucose from gummy jelly [34, 35]. Accordingly, to answer our clinical question, we designed a cross-sectional study to measure masticatory efficiency quantitatively by a new testing system and assess microvascular complications in patients with diabetes.

## 2. Materials and methods

### 2.1 Subjects and study design

Participants of this study were enrolled from patients with type 2 diabetes who underwent educational hospitalization in the Department of Endocrinology and Diabetic Medicine, Hiroshima University Hospital, from April 2016 to March 2020. All participants in this study provided written informed consent prior to enrollment, consistent with the Helsinki Declaration and guidelines of the institutional review board of Hiroshima University. Masticatory efficiency was measured by using a gummy jelly in the Center for Oral Clinical Examination, Hiroshima University, and periodontitis was assessed in the Department of Periodontics, Hiroshima University Hospital. This study was approved by the Ethical Committee for Epidemiology of Hiroshima University at November 19, 2015. The approval number was E-150.

### 2.2 Assessment of diabetes

All the subjects received biochemical evaluations after blood sample collection. Plasma HbA1c was assessed by high-performance liquid chromatography. Serum creatinine (Cre) levels were measured by clinical analyzers. The estimated glomerular filtration rate (eGFR) was calculated according to age and Cre as follows: men; $194 \times Cre^{-1.094} \times age^{-0.287}$, women; $194 \times Cre^{-1.094} \times age^{-0.287} \times 0.739$ [36]. Diabetic nephropathy was defined based on the clinical practice guidelines of the Japan Diabetes Society[Stage 2: UAE 30–299 mg/day (or ACR 30–299 mg/g Cre), Stage 3: UAE ≥300 mg/day (or ACR ≥300 mg/g Cre) or a protein creatinine ratio ≥0.5 g/g Cre, Stage 4: eGFR <30 mL/min/1.73 m$^2$, and Stage 5: maintenance dialysis] [37]. Daily urinary albumin excretion (UAE) and the albumin–creatinine ratio (ACR) were measured by immunoturbidimetric assays from 24-hour urine samples. Diabetic neuropathy was defined by the presence of at least 2 of the 3 following factors: 1) subjective symptoms, 2) reduced Achilles tendon reflex, or 3) decreased vibration sensation on the internal malleolus. These abbreviated diagnostic criteria, which are frequently employed for the diagnosis in Japan, are commended by Diabetic Neuropathy Study Group in Japan because of their sensitivity (68%) and specificity (74%) obtained by evaluating nerve conduction study as a gold standard [38]. The presence of diabetic retinopathy was diagnosed by ophthalmologists.

### 2.3 Assessment of periodontitis

The periodontal inflammatory surface area (PISA) is an index to assess the overall periodontal inflammatory status [39]. The PISA enables a quantitative evaluation of the burden of periodontal inflammation as a continuous variable in individual patients. To calculate the PISA, all participants were assessed for probing pocket depth and bleeding on probing at 6 sites per tooth, and these values were entered into a spreadsheet that is freely accessible at http://www.parsprototo.info/docs/PISA_CAL.xls [39]. The number of moving teeth was also assessed using Miller's mobility index, which is the most widely accepted method for the routine clinical examination of tooth mobility; class 0 represents no mobility, and classes 1–3 represent consecutive degrees of tooth mobility [40].

## 2.4 Measurement of masticatory efficiency

Masticatory efficiency was measured by using a masticatory ability testing system (GLUCO SENSOR GS-II, GC Corporation, Tokyo, Japan). In brief, patients freely chewed 2 g of gummy jelly, which included approximately 100 mg of glucose, measuring approximately 14 mm (diameter) × 10 mm (height) in size and weighing approximately 2 g (Gurukoramu, GC Corporation) for 20 s. Subsequently, the patients rinsed with 10 mL of water and spat both the chewed gummy jelly and the water into a cup through a 2 mm mesh. The amount of eluted glucose in the filtrate was measured using a glucose measuring device (GLUCO SENSOR GS-II). The procedure was performed twice, and the average value of the results was defined as the quantitative parameter of masticatory efficiency [34, 35]. The patient using removable prostheses were assessed with them in place.

## 2.5 Statistical analysis

Statistical analyses were performed using JMP® 13 (SAS Institute Inc., Cary, NC, USA). Correlations between continuous variables [i.e., age; body mass index (BMI); duration of diabetes; HbA1c; number of remaining teeth; number of moving teeth; PISA] and masticatory efficiency were assessed by Spearman rank correlation coefficient. The effects of ordinal variables (i.e., the stage of diabetic nephropathy) or nominal variables (i.e., sex; current smoking; presence of diabetic neuropathy or diabetic retinopathy) on masticatory efficiency were also assessed by one-way ANOVA or the t-test. Multivariable linear regression models were constructed to examine which factors were related to masticatory efficiency. The average amount of eluted glucose was defined as an objective variable. The covariates were included the factor of interest (i.e., sex; age; duration of diabetes; the presence of diabetic neuropathy and retinopathy; the stage of diabetic nephropathy; the number of remaining teeth; the number of moving teeth; PISA). Here, there is a fact that oral function, including masticatory efficiency, is apparently decreased in a patient whose number of remaining teeth is less than 20 [41]. Thus, to avoid a skewed assessment by subjects with reduced teeth number, we stratified patients who had more than 20 teeth for the statistical analysis. More specifically, the subjects with lower than 20 teeth were eliminated even if removal prostheses were placed. Statistical significance was defined as a two-sided $p$ value of $< 0.05$.

## 3. Results

### 3.1 Baseline characteristics

The participant characteristics are listed in Table 1. This cross-sectional study included 172 individuals who agreed to participate in this study, comprising 100 men and 72 women and their mean age was 61 ± 14. Their mean duration of diabetes was 11.1 ± 10.7 years and their mean HbA1c was 10.3 ± 2.1% (87 ± 21 mmol/mol). Among the diabetic complications, 122 patients complicated with diabetic neuropathy and 43 patients complicated with diabetic retinopathy. Diabetic nephropathy consisted of 91 patients with stage 1, 59 patients with stage2, 12 patients with stage 3 and 10 patients with stage 4. There were no patients with stage 5 of diabetic nephropathy. The removable prostheses were placed in 46 patients and no patients were treated with dental implants. The mean masticatory efficiency assessed by the amount of eluted glucose was 172.4 ± 60.0 mg/dL.

### 3.2 Correlation between masticatory efficiency and diabetic microvascular complications

We first performed bivariable analyses to assess associations between participant characteristics and masticatory efficiency (Fig 1). Masticatory efficiency was significantly negatively

**Table 1. Participant characteristics at baseline.**

| Parameter | Value |
|---|---|
| *n* | 172 |
| Male/female, *n* (%) | 100 (58)/72 (42) |
| Age (years) | 60.8 ± 15.4 |
| BMI (kg/m$^2$) | 25.7 ± 6.2 |
| Current smoking, *n* (%) | 37 (22) |
| Duration of diabetes (years) | 11.1 ± 10.7 |
| Systolic blood pressure (mmHg) | 129 ± 19 |
| Diastolic blood pressure (mmHg) | 78 ± 13 |
| HbA1c [% (mmol/mol)] | 10.3 ± 2.1 (87 ± 21) |
| UAE (mg/day) | 20.6 (11.4, 60.6) |
| eGFR (ml/min/1.73 m$^2$) | 74.7 ± 30.3 |
| Diabetic neuropathy, *n* (%) | 122 (71) |
| Diabetic retinopathy, *n* (%) | 43 (25) |
| Diabetic nephropathy, *n* (%) | |
| Stage 1 | 91 (53) |
| Stage 2 | 59 (34) |
| Stage 3 | 12 (7) |
| Stage 4 | 10 (6) |
| Stage 5 | 0 (0) |
| Number of remaining teeth, *n* | 23 ± 7 |
| Number of moving teeth, *n* | 2 ± 3 |
| PISA (mm$^2$) | 539.4 ± 484.6 |
| Masticatory efficiency (mg/dL) | 172.4 ± 60.0 |

UAE is expressed as median (first quartile, third quartile). Other values are expressed as the mean ± standard deviation or n (%). Abbreviations: BMI, body mass index; HbA1c, hemoglobin A1c; UAE, daily urinary albumin excretion; eGFR, estimated glomerular filtration rate; PISA, periodontal inflammatory surface area.

correlated with duration of diabetes ($p$ = 0.049; Fig 1C) and the number of moving teeth ($p$ = 0.007; Fig 1F) and positively correlated with the number of remaining teeth ($p$ < 0.0001; Fig 1E), whereas age, BMI, HbA1c and PISA did not have significant associations with masticatory efficiency (Fig 1A, 1B, 1D and 1G). Then, we investigated the correlation between masticatory efficiency and diabetic conditions, including neuropathy, retinopathy, and nephropathy (Fig 2). Masticatory efficiency was significantly reduced in patients with diabetic neuropathy ($p$ < 0.0001; Fig 2A). On the other hand, diabetic nephropathy and retinopathy were not associated with the level of masticatory efficiency (Fig 2B and 2C). Moreover, sex and current smoking were not correlated with masticatory efficiency (Fig 2D and 2E).

In the multivariable linear regression analysis, masticatory efficiency was positively associated with the number of remaining teeth ($p$ < 0.0001) and negatively associated with the complication of diabetic neuropathy ($p$ = 0.007) (Table 2). In contrast, duration of diabetes and the number of moving teeth, which were statistically significant in the bivariable analysis, was not associated with masticatory efficiency in the multivariable analysis (Table 2).

### 3.3 Stratified analysis of the patients with more than 20 remaining teeth

Since the remaining number of teeth less than 20 reduces the oral function [41], in order to avoid a skewed assessment, we stratified the patients with more than 20 remaining teeth. As a result, 124 patients had more than 20 remaining teeth, of which 12 patients used removable

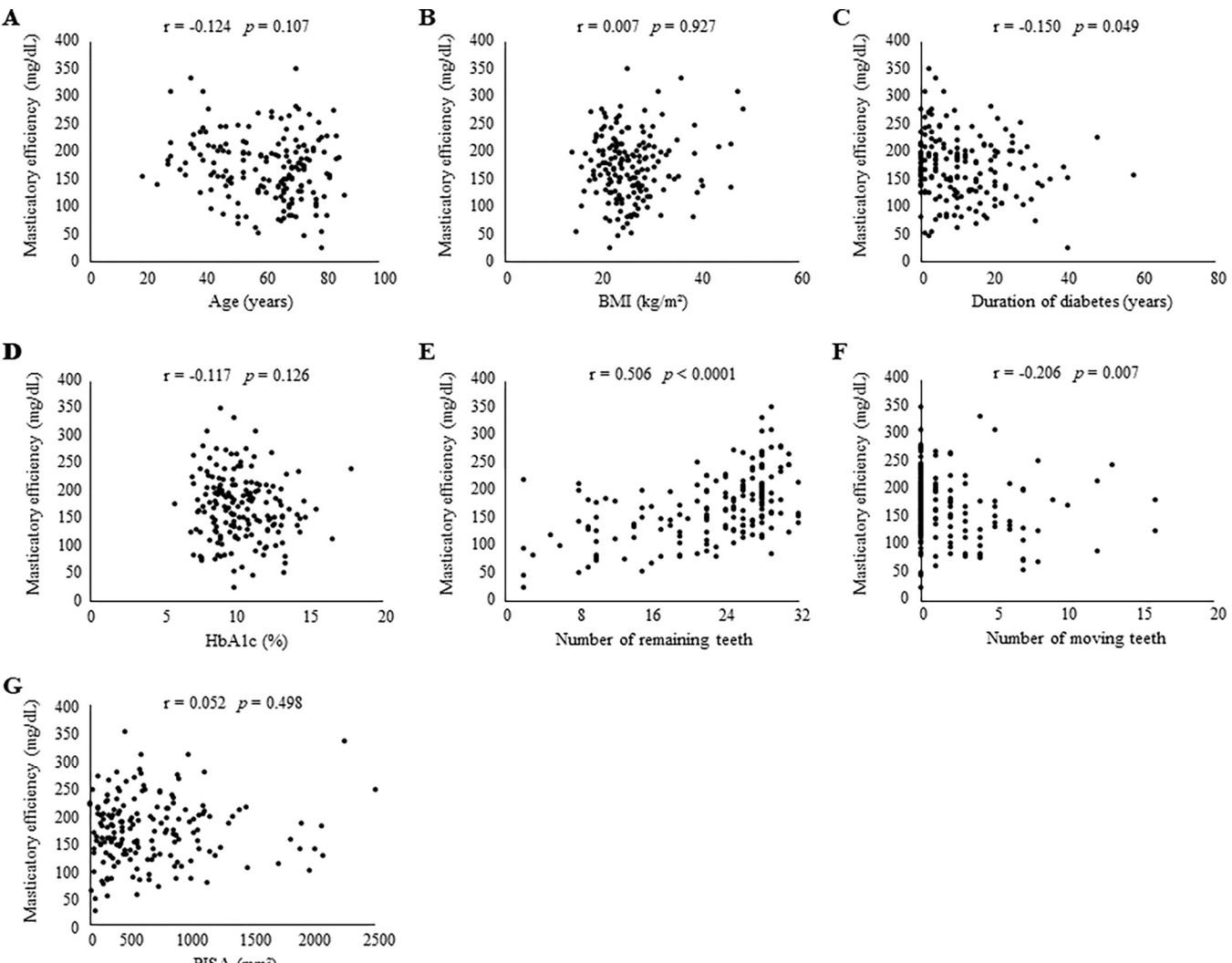

**Fig 1. Spearman rank correlation.** Correlations between masticatory efficiency and (A) age, (B) BMI, (C) duration of diabetes, (D) HbA1c, (E) number of remaining teeth, (F) number of moving teeth and (G) PISA were assessed. Spearman rank correlation coefficients (r) and $p$ values ($p$) are shown. Statistical significance was defined as a two-sided $p$ value of $< 0.05$. Abbreviations: BMI, body mass index; HbA1c, hemoglobin A1c; PISA, periodontal inflammatory surface area.

prostheses. The presence of diabetic neuropathy did not affect the number of remaining teeth (data not shown). Nevertheless, the complication of diabetic neuropathy showed lower mastication (Fig 3). Furthermore, the multivariable linear regression analysis indicated a significant correlation between masticatory efficiency and the number of remaining teeth ($p = 0.0002$); diabetic neuropathy ($p = 0.0004$) (Table 3).

## 4. Discussion

The present study revealed, for the first time, that patients with type 2 diabetes who have diabetic neuropathy have significantly lower masticatory efficiency. Besides, stratified analysis for the patients with more than 20 remaining teeth demonstrated that the negative effect of diabetic neuropathy remained. Taken together, these findings suggested the association between masticatory efficiency and diabetic neuropathy. Since masticatory dysfunction is an obstacle to

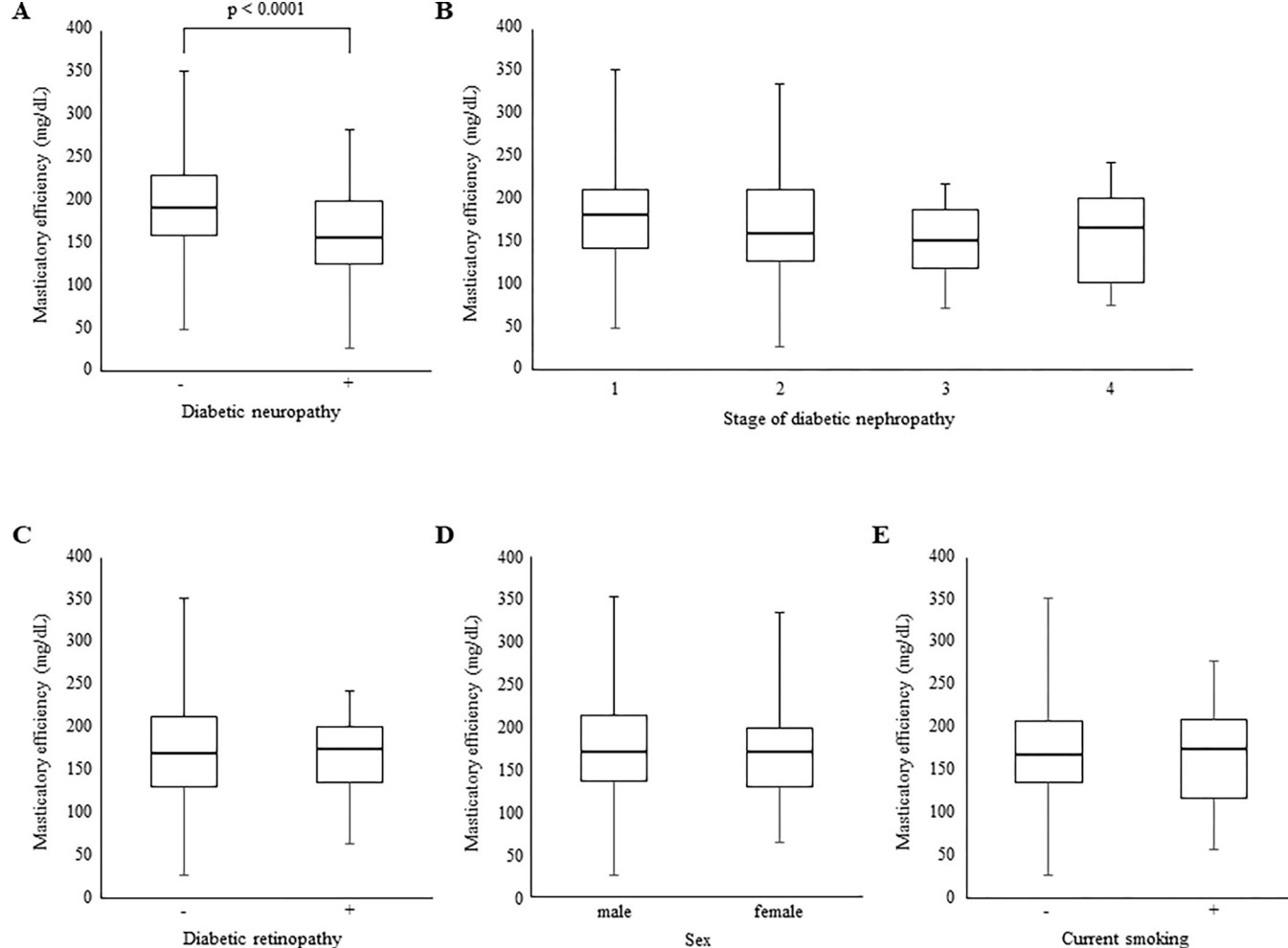

**Fig 2. One-way ANOVA and the t-test.** Effects of (A) diabetic neuropathy, (B) stage of diabetic nephropathy, (C) diabetic retinopathy, (D) sex and (E) current smoking on masticatory efficiency were assessed. Statistical significance was defined as a two-sided p value of < 0.05.

medical diet therapy, masticatory efficiency in diabetic patients with diabetic neuropathy may be a useful clinical indicator for successful diabetes therapy. On the other hand, unexpectedly, neither diabetic retinopathy nor diabetic nephropathy correlated with masticatory efficiency. These results imply that diabetic neuropathy could be the cause of the resultant masticatory disorder in diabetes patients. We discuss the plausible mechanisms of impairment for mastication below.

First, the possible factor responsible for masticatory dysfunction in diabetic neuropathy patients may be sarcopenia. It is widely accepted that type 2 diabetes is associated with sarcopenia [42–44]. Furthermore, diabetic neuropathy is one of the important risk factors for the occurrence of sarcopenia in diabetic patients [45–47]. Notably, sarcopenia induces masseter muscle atrophy and chewing dysfunction [48, 49]. Although this present clinical study did not assess the motor neuronal function, diabetic neuropathy was correlated with masticatory disorder. Consequently, it is plausible that sarcopenia associated with diabetic neuropathy may cause masticatory muscle mass reduction, which results in the development of masticatory dysfunction. Since this present clinical study lacked the data evaluating sarcopenia, including such as muscle strength, physical function and muscle mass, we cannot show the precise

**Table 2. Multivariable analysis.**

| Parameter | 95% CI, upper/lower | *p* value |
|---|---|---|
| Sex | 2.672/-13.418 | 0.189 |
| Age | 0.962/-0.221 | 0.218 |
| Duration of diabetes | 0.716/-1.075 | 0.693 |
| Diabetic neuropathy | 21.522/3.476 | 0.007 |
| Diabetic retinopathy | 10.041/-10.111 | 0.995 |
| Diabetic nephropathy | | |
| Stage 2–1 | 14.161/-20.341 | 0.724 |
| Stage 3–2 | 25.404/-40.981 | 0.644 |
| Stage 4–3 | 41.145/-47.214 | 0.892 |
| Number of remaining teeth | 5.384/2.984 | < 0.0001 |
| Number of moving teeth | 2.601/-3.297 | 0.816 |
| PISA | 0.009/-0.029 | 0.308 |

Statistical significance was defined as a two-sided *p* value of < 0.05. Abbreviations: CI, confidence interval; PISA, periodontal inflammatory surface area.

association of sarcopenia for masticatory disorder in patients with diabetic neuropathy. To clarify this tentative hypothesis, additional studies evaluating the relation between sarcopenia/ diabetic motor neuropathy and masticatory dysfunction will be needed. More specifically, in addition to our mastication efficacy testing system, recording the mandibular kinematics and muscular activation during chewing will strengthen such promising future study [50].

The other plausible factor of impairment for mastication could be trigeminal nerve dysfunction due to diabetic neuropathy. Although most of the facial muscles are innervated by the facial nerve, the muscles involved in mastication, consisting of the masseter, temporal, medial pterygoid, and lateral pterygoid muscles, are innervated by the trigeminal nerve. When these mandibular motor fibers are affected, the patients may complain of weakness in chewing [51]. Thus, masticatory dysfunction in the patients with diabetic neuropathy in this study may be attributed to impaired trigeminal nerve function. Oculomotor, abducens, facial, and trochlear nerve palsies are frequently observed as diabetic mononeuropathies; however, there are no reports demonstrating trigeminal nerve dysfunction in patients with diabetic neuropathy. However, pure trigeminal motor neuropathy, caused by viral or bacterial infection [52, 53], tumors [54] or unknown causes [55], can lead to masticatory muscle dysfunction. Accordingly, it is highly conceivable that masticatory dysfunction due to trigeminal nerve palsy occurs in diabetic neuropathy. More importantly, compared to the well-reported prognosis of diabetic mononeuropathy, which can be treated by controlling blood glucose levels, the prognosis of trigeminal nerve palsy may be relatively poor. As described above, masticatory disorder caused by trigeminal nerve dysfunction may disrupt healthy eating habits and thereby worsen blood glucose control.

PISA was not associated with masticatory dysfunction in this study; however, previous studies have demonstrated a correlation between periodontitis and masticatory disability [10–12]. Despite this, our results may not contradict those of previous reports. The PISA reflects the inflamed surface area of periodontal pockets. In brief, as the number of remaining teeth decreased, the PISA decreased. Accordingly, severe periodontitis patients who lose their teeth and develop masticatory disorder have a decreased PISA.

In this study, diabetic neuropathy was diagnosed by abbreviated diagnostic criteria, which are generally employed in Japan. These abbreviated diagnostic criteria are easy and reliable for

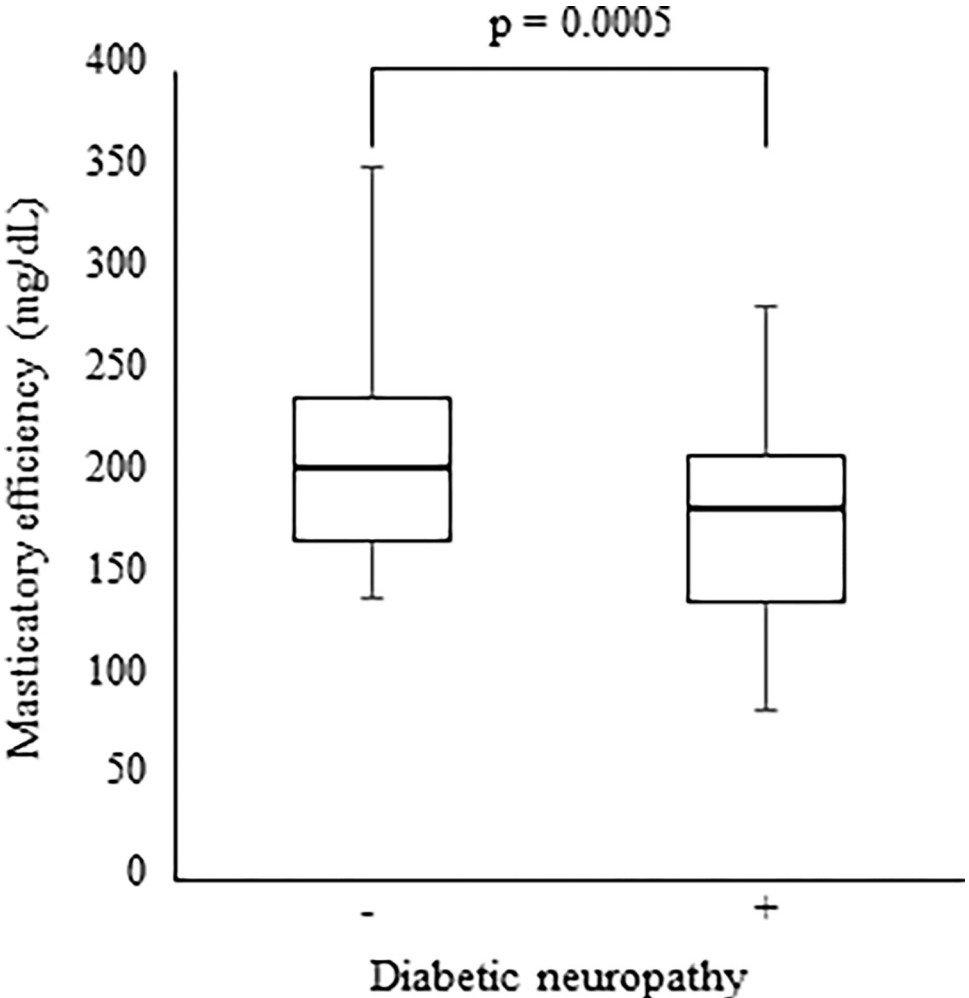

**Fig 3. Effect of diabetic neuropathy on masticatory efficiency in patients with more than 20 remaining teeth.**
Statistical significance was defined as a two-sided *p* value of < 0.05.

the diagnosis but not quantitative. Thus, to investigate the association between the objective degree of diabetic neuropathy and masticatory disorder, a quantitative assessment such as a nerve conduction test will be required. Moreover, a future study testing the association between diabetic neuropathy and neuromuscular control of mastication will clarify the effect of diabetic neuropathy on masticatory dysfunction in more detail. To this end, in addition to the masticatory efficiency testing system, monitoring of the mandibular kinematic and muscular activity should be employed.

## 5. Conclusion

In conclusion, we revealed a significant association between diabetic neuropathy and masticatory dysfunction. Additional assessment for a mandibular kinematic and muscular activity will deepen our understanding of the neuropathic effects of diabetes on the craniofacial disorder.

For diabetic patients with masticatory disorder, it may be difficult to implement a dietary treatment regimen. To control hyperglycemic conditions in diabetic neuropathy patients, diet therapy considering a patient's masticatory function by dentists and nutritionists could be helpful.

**Table 3. Multivariable analysis after stratification.**

| Parameter | 95% CI, upper/lower | *p* value |
|---|---|---|
| Sex | 4.840/-13.596 | 0.349 |
| Age | 1.288/-0.079 | 0.082 |
| Duration of diabetes | 1.106/-1.371 | 0.832 |
| Diabetic neuropathy | 28.792/8.606 | 0.0004 |
| Diabetic retinopathy | 18.473/-6.202 | 0.327 |
| Diabetic nephropathy | | |
| Stage 2–1 | 26.450/-14.048 | 0.545 |
| Stage 3–2 | 45.093/-36.560 | 0.836 |
| Stage 4–3 | 49.857/-60.392 | 0.850 |
| Number of remaining teeth | 10.646/3.396 | 0.0002 |
| Number of moving teeth | 4.172/-3.501 | 0.170 |
| PISA | 0.010/-0.031 | 0.291 |

Statistical significance was defined as a two-sided *p* value of < 0.05. Abbreviations: CI, confidence interval; PISA, periodontal inflammatory surface area.

## Supporting information

**S1 File.**
(XLSX)

## Acknowledgments

The authors gratefully acknowledge the dedication of the patients who volunteered to participate in this study. The authors also thank the dental hygienists who were responsible for the patients' oral health education: C. Kozono, Y. Hamamoto, and Y. Mizota (Dental Section, Department of Clinical Support, Hiroshima University Hospital, Hiroshima, Japan).

## Author Contributions

**Conceptualization:** Yuta Hamamoto, Kazuhisa Ouhara, Nao Komatsu, Mikihito Kajiya, Shinji Matsuda, Tsuyoshi Fujita, Shinya Sasaki, Haruya Ohno, Masayasu Yoneda, Hidemi Kurihara.

**Data curation:** Yuta Hamamoto, Tsuyoshi Miyagawa, Tomoaki Shintani, Tomoyuki Iwata.

**Formal analysis:** Yuta Hamamoto, Tsuyoshi Miyagawa, Tomoaki Shintani, Shinji Matsuda.

**Funding acquisition:** Yuta Hamamoto, Kazuhisa Ouhara, Tsuyoshi Miyagawa.

**Investigation:** Yuta Hamamoto, Kazuhisa Ouhara, Tomoaki Shintani, Nao Komatsu, Mikihito Kajiya, Shinji Matsuda, Shinya Sasaki, Tomoyuki Iwata.

**Methodology:** Yuta Hamamoto, Tsuyoshi Miyagawa, Mikihito Kajiya, Haruya Ohno.

**Project administration:** Yuta Hamamoto, Masayasu Yoneda, Noriyoshi Mizuno, Hidemi Kurihara.

**Resources:** Tomoaki Shintani, Haruya Ohno, Masayasu Yoneda.

**Supervision:** Yuta Hamamoto, Kazuhisa Ouhara, Masayasu Yoneda, Noriyoshi Mizuno, Hidemi Kurihara.

**Validation:** Yuta Hamamoto, Tsuyoshi Miyagawa, Mikihito Kajiya.

**Visualization:** Yuta Hamamoto.

**Writing – original draft:** Yuta Hamamoto, Mikihito Kajiya.

**Writing – review & editing:** Yuta Hamamoto, Mikihito Kajiya, Shinji Matsuda, Haruya Ohno, Masayasu Yoneda.

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
