## [Decision Letter · Decision Letter 0]

27 Feb 2022

PONE-D-22-02324Masticatory dysfunction in patients with diabetic neuropathy: A cross-sectional studyPLOS ONE

Dear Dr. Hamamoto,

Thank you for submitting your manuscript to PLOS ONE. After careful consideration, we feel that it has merit but does not fully meet PLOS ONE’s publication criteria as it currently stands. Therefore, we invite you to submit a revised version of the manuscript that addresses the points raised during the review process.

We look forward to receiving your revised manuscript.

Kind regards,

Kanhaiya Singh, Ph.D

Academic Editor

PLOS ONE

Journal Requirements:

Additional Editor Comments:

Although Reviewers have found merit in this manuscript, they have recommended significant revision. Please address the comments about quantitative assessment and possible selection bias in the study.

Reviewers' comments:

Reviewer's Responses to Questions

**Comments to the Author**

1. Is the manuscript technically sound, and do the data support the conclusions?

Reviewer #1: No

Reviewer #2: Yes

2. Has the statistical analysis been performed appropriately and rigorously? 

Reviewer #1: Yes

Reviewer #2: Yes

3. Have the authors made all data underlying the findings in their manuscript fully available?

Reviewer #1: Yes

Reviewer #2: Yes

4. Is the manuscript presented in an intelligible fashion and written in standard English?

Reviewer #1: No

Reviewer #2: Yes

5. Review Comments to the Author

Reviewer #1: While the paper entitled “Masticatory dysfunction in patients with diabetic neuropathy” is interesting and insightful, some revisions of the text are needed.

Materials and Methods: since the paper focuses on diabetic neuropathy, the fact that it was exclusively diagnosed by means of clinical examination, without the use of any of the available quantitative tests, is problematic. Moreover, it precludes the possibility to investigate the association between the objective degree of neuropathy and the reduction in masticatory efficiency.

Patient description is also problematic: a major bias is due to the fact that no mention is made of the occlusal condition of the patients the presence of prosthodontic rehabilitation, which is very likely to be present given the age and condition of the subjects, and which is certainly capable of altering masticatory efficiency. No information is given as to whether the masticatory efficiency test was carried out with or without removable prostheses in place. Moreover, the presence of fixed prosthodontic rehabilitation, either on natural teeth or implants, should be noted and taken into account. No attempt was made to stratify the subjects according to the number of teeth still present, even though it is well attested that masticatory efficiency sharply decreases if the number of remaining teeth decreases below 20. In the context of this research, a few subject with an extremely compromised dentition may have skewed the analysis exaggerating the effect of tooth loss. At the very least, the presence of prosthodontic rehabilitation should be discussed.

Results: the meaning of line 160-161 is unclear. Line 183-184 is unclear in the context as well, and needs to be clarified (positively or negatively associated?). the same goes for line 210 (…were associated with reduced masticatory efficiency).

Discussion: in the context of this paper’s research, to hypothesize a role for diabetic neuropathy in masticatory dysfunction is not of itself implausible (although it should probably be quite advanced to manifest on the trigeminal nerve), but it really is not well tested in the context of this research. The only way to do it is to separate the confounding factor or remaining teeth (a bias in this case) and investigate the effects of neuropathy on neuromuscular control of mastication, not on masticatory efficiency. This distinction, that admittedly reduces the claim of the paper, should be included and discussed.

Reviewer #2: The manuscript is well written, the statistical data are satisfactory and self explanatory. The figures provided in the manuscript lacks visual clarity and can be made more prominent. There are some background correction to be made in each figure for clarity. Thanks

6. PLOS authors have the option to publish the peer review history of their article (what does this mean?). If published, this will include your full peer review and any attached files.

Reviewer #1: **Yes: **Maria Grazia Piancino

Reviewer #2: No

---

## [Author Response · Author response to Decision Letter 0]

13 Apr 2022

Response to Reviewer Comments

Reviewer #1

1. While the paper entitled “Masticatory dysfunction in patients with diabetic neuropathy” is interesting and insightful, some revisions of the text are needed.

Response: We are delighted to hear that the reviewer thinks our manuscript is interesting, albeit with some drawbacks. To increase the quality of our manuscript, we followed the suggestions described below. 

2. Materials and Methods: since the paper focuses on diabetic neuropathy, the fact that it was exclusively diagnosed by means of clinical examination, without the use of any of the available quantitative tests, is problematic. Moreover, it precludes the possibility to investigate the association between the objective degree of neuropathy and the reduction in masticatory efficiency.

Response: We completely agree with this reviewer’s insightful criticism. In this present study, diabetic neuropathy was diagnosed using the abbreviated diagnostic criteria. These criteria are frequently employed for the diagnosis in Japan and are commended by Diabetic Neuropathy Study Group in Japan because of their sensitivity (68%) and specificity (74%) obtained by evaluating nerve conduction test as a gold standard (Yasuda H et al., Diabetes Res Clin Pract, 2007). Thus, we believe our present study using the abbreviated diagnostic criteria can support our conclusion. However, as the reviewer pointed out, the quantitative analysis such as the nerve conduction test should be helpful to assess the association between the objective degree of neuropathy and the reduction in masticatory efficiency. This point is discussed in the revised discussion section.

Materials and Methods section: Page 6, line 118-121

Discussion section: Page 14, line 285-289

3. Patient description is also problematic: a major bias is due to the fact that no mention is made of the occlusal condition of the patients the presence of prosthodontic rehabilitation, which is very likely to be present given the age and condition of the subjects, and which is certainly capable of altering masticatory efficiency. No information is given as to whether the masticatory efficiency test was carried out with or without removable prostheses in place. Moreover, the presence of fixed prosthodontic rehabilitation, either on natural teeth or implants, should be noted and taken into account. No attempt was made to stratify the subjects according to the number of teeth still present, even though it is well attested that masticatory efficiency sharply decreases if the number of remaining teeth decreases below 20. In the context of this research, a few subject with an extremely compromised dentition may have skewed the analysis exaggerating the effect of tooth loss. At the very least, the presence of prosthodontic rehabilitation should be discussed.

Response: We sincerely appreciate this reviewer’s keen eye. Prosthodontic rehabilitation and the number of remaining teeth should be considered in more detail. First of all, the removable protheses were used in 46 patients, whereas no dental implant was observed in all subjects. Besides, the measurement of masticatory efficiency was conducted with the removable prostheses in place. These points are described in the revised Materials and Methods and Results section.

In addition, as the reviewer pointed out, it is true that the reduced number of remaining teeth, less than 20, should affect oral function, including masticatory efficiency. Including such subjects may have skewed our analysis. Accordingly, encouraged by this reviewer’s constructive comment, we conducted a stratified analysis of the patients with more than 20 remaining teeth. As a result, we found that diabetic neuropathy decreased mastication efficacy. This finding can support our conclusion that the complication of diabetic neuropathy is associated with masticatory disfunction. 

Materials and Methods section: Page 7, line 143-144

Page 7, line 158-162

Results section: Page 8, line 173-174

Page 11, line 223- Page 12, line 235

New Figure 3

New Table 3

Discussion section: Page 13, line 245-248

4. Results: the meaning of line 160-161 is unclear. Line 183-184 is unclear in the context as well, and needs to be clarified (positively or negatively associated?). the same goes for line 210 (…were associated with reduced masticatory efficiency).

Response: In accordance with the reviewer’s suggestion, we have amended these unreadable sentences.

Results section: Page 8, line 169-171

Page 10, line 195

Page 10, line 211-213

5. Discussion: in the context of this paper’s research, to hypothesize a role for diabetic neuropathy in masticatory dysfunction is not of itself implausible (although it should probably be quite advanced to manifest on the trigeminal nerve), but it really is not well tested in the context of this research. The only way to do it is to separate the confounding factor or remaining teeth (a bias in this case) and investigate the effects of neuropathy on neuromuscular control of mastication, not on masticatory efficiency. This distinction, that admittedly reduces the claim of the paper, should be included and discussed.

Response: We sincerely appreciate this insightful suggestion. We agree entirely that this present study is still enough to draw the precise association between diabetic neuropathy and masticatory function, even though a new stratified analysis can solve the remaining teeth number problems. Following the reviewer’s constructive comments, we will conduct the next study assessing the neuromuscular control of mastication. This point regarding the future study is discussed in the revised Discussion section.

Discussion section: Page 14, line 289-291

Reviewer #2

1. The manuscript is well written, the statistical data are satisfactory and self explanatory. The figures provided in the manuscript lacks visual clarity and can be made more prominent. There are some background correction to be made in each figure for clarity. Thanks.

Response: We sincerely appreciate these positive and encouraging comments on our manuscript. Following this reviewer’s constructive suggestion, we have amended the incorrect notation in Figure 1. Besides, Figures 1 and 2 are revised to be more visible. 

Revised Figure1 and 2

---

## [Decision Letter · Decision Letter 1]

28 Apr 2022

PONE-D-22-02324R1Masticatory dysfunction in patients with diabetic neuropathy: A cross-sectional studyPLOS ONE

Dear Dr. Hamamoto,

Thank you for submitting your manuscript to PLOS ONE. After careful consideration, we feel that it has merit but does not fully meet PLOS ONE’s publication criteria as it currently stands. Therefore, we invite you to submit a revised version of the manuscript that addresses the points raised during the review process.

We look forward to receiving your revised manuscript.

Kind regards,

Kanhaiya Singh, Ph.D

Academic Editor

PLOS ONE

Additional Editor Comments (if provided):

Please address to the additional comments made by Reviewer 1.

Reviewers' comments:

Reviewer's Responses to Questions

**Comments to the Author**

1. If the authors have adequately addressed your comments raised in a previous round of review and you feel that this manuscript is now acceptable for publication, you may indicate that here to bypass the “Comments to the Author” section, enter your conflict of interest statement in the “Confidential to Editor” section, and submit your "Accept" recommendation.

Reviewer #1: (No Response)

2. Is the manuscript technically sound, and do the data support the conclusions?

Reviewer #1: Partly

3. Has the statistical analysis been performed appropriately and rigorously? 

Reviewer #1: Yes

4. Have the authors made all data underlying the findings in their manuscript fully available?

Reviewer #1: No

5. Is the manuscript presented in an intelligible fashion and written in standard English?

Reviewer #1: Yes

6. Review Comments to the Author

Reviewer #1: The revised manuscript has been improved, but some points are still lacking. Considering the significant limitations of the masticatory ability testing system as methodology for the evaluation of a complex pathology such as diabetes in this study, this paper needs to be careful about the strength of the conclusions. Association does not imply a cause/effect relationship, and demonstrating one will require experimental evidence, which is, by its very design, not presented by the study. Nevertheless, it is useful and inherently interesting to speculate on the possible mechanism linking diabetes and reduced mastication, given the high prevalence of both conditions and the relevance of the topic.

Introduction

Please mention the actual possibility of recording the mandibular kinematic of the chewing patterns during chewing that gives information on the peripheral effects of neuromuscular motor control that would be useful for understanding the neuropathic effects of diabetes(doi: 10.1093/ejo/cjr109.).

Discussion

Line 267 please add that to properly consider the role of sarcopenia during mastication it is necessary to record the mandibular kinematics and muscular activation during chewing (doi: 10.1016/j.archoralbio.2016.03.013.). To this end methods other than masticatory ability testing system are needed.

Please add at the end of the discussion as limitation of the study that the methodology used i.e. the ability testing system, is not refined enough for the study of a complex pathology such as diabetes and that the future direction will be the recording of the mandibular kinematic together with the muscular activation to establish a deeper understanding of the topic.

Abstract and Conclusion

Please, add a reference to the methodological limitation of the study.

7. PLOS authors have the option to publish the peer review history of their article (what does this mean?). If published, this will include your full peer review and any attached files.

Reviewer #1: No

---

## [Author Response · Author response to Decision Letter 1]

12 May 2022

Response to Reviewer Comments

Reviewer #1

1. The revised manuscript has been improved, but some points are still lacking. Considering the significant limitations of the masticatory ability testing system as methodology for the evaluation of a complex pathology such as diabetes in this study, this paper needs to be careful about the strength of the conclusions. Association does not imply a cause/effect relationship, and demonstrating one will require experimental evidence, which is, by its very design, not presented by the study. Nevertheless, it is useful and inherently interesting to speculate on the possible mechanism linking diabetes and reduced mastication, given the high prevalence of both conditions and the relevance of the topic.

Response: We appreciate this reviewer #1, who acknowledged that our first revision responding to the previous questions was improved and our study could be potentially interesting. Besides, we agree with the new critiques suggesting the effectiveness of the other examinations. To increase the quality of our manuscript, we followed such constructive suggestions described below. 

2. Introduction: Please mention the actual possibility of recording the mandibular kinematic of the chewing patterns during chewing that gives information on the peripheral effects of neuromuscular motor control that would be useful for understanding the neuropathic effects of diabetes(doi: 10.1093/ejo/cjr109.).

Response: We completely agree that assessing the mandibular kinematic is helpful in clarifying the neuromuscular motor function. Thus, this point was mentioned in the revised introduction section with appropriate reference (Piancino et al., 2021, European Journal of Orthodontics). 

Introduction section: Page 3 line 85-87

3. Discussion: Line 267 please add that to properly consider the role of sarcopenia during mastication it is necessary to record the mandibular kinematics and muscular activation during chewing (doi: 10.1016/j.archoralbio.2016.03.013.). To this end methods other than masticatory ability testing system are needed.

Please add at the end of the discussion as limitation of the study that the methodology used i.e. the ability testing system, is not refined enough for the study of a complex pathology such as diabetes and that the future direction will be the recording of the mandibular kinematic together with the muscular activation to establish a deeper understanding of the topic.

Response: We sincerely appreciate this insightful comment. Monitoring the mandibular kinematics and muscular activation should be helpful in clarifying the relationship between sarcopenia and masticatory function. Following this reviewer’s suggestion, we introduced this idea with a proper reference (Piancino et al., 2016, Archives of Oral Biology) in the revised manuscript. Besides, at the end of the discussion section, we have referred that the additional examinations recording mandibular kinematic and muscular activity will strengthen our findings in this present study.

Discussion section: Page 13, line 268- Page 14, line 271. Page 15, line 301-303

4. Abstract and Conclusion: Please, add a reference to the methodological limitation of the study.

Response: In accordance with the reviewer’s comment, we have referred to the necessity of the mandibular kinetics and muscle activity examinations in the revised conclusion section. On the other hand, the abstract section, having a character limit, simply referred to the results demonstrating that diabetic patients who developed diabetic neuropathy had reduced masticatory efficiency. Thus, we think the abstract in this paper may not need to be amended from its current form. 

Conclusion section: Page 16, line 307-309.

---

## [Decision Letter · Decision Letter 2]

25 May 2022

Masticatory dysfunction in patients with diabetic neuropathy: A cross-sectional study

PONE-D-22-02324R2

Dear Dr. Hamamoto,

We’re pleased to inform you that your manuscript has been judged scientifically suitable for publication and will be formally accepted for publication once it meets all outstanding technical requirements.

Kind regards,

Kanhaiya Singh, Ph.D

Academic Editor

PLOS ONE

Additional Editor Comments (optional):

Reviewers' comments:

Reviewer's Responses to Questions

**Comments to the Author**

1. If the authors have adequately addressed your comments raised in a previous round of review and you feel that this manuscript is now acceptable for publication, you may indicate that here to bypass the “Comments to the Author” section, enter your conflict of interest statement in the “Confidential to Editor” section, and submit your "Accept" recommendation.

Reviewer #1: All comments have been addressed

2. Is the manuscript technically sound, and do the data support the conclusions?

Reviewer #1: Yes

3. Has the statistical analysis been performed appropriately and rigorously? 

Reviewer #1: Yes

4. Have the authors made all data underlying the findings in their manuscript fully available?

Reviewer #1: Yes

5. Is the manuscript presented in an intelligible fashion and written in standard English?

Reviewer #1: Yes

6. Review Comments to the Author

Reviewer #1: The comments have been correctly addressed, the manuscript has been improved and is now ready to be published.

7. PLOS authors have the option to publish the peer review history of their article (what does this mean?). If published, this will include your full peer review and any attached files.

Reviewer #1: No

---

## [Editor Report · Acceptance letter]

27 May 2022

PONE-D-22-02324R2 

Masticatory dysfunction in patients with diabetic neuropathy: A cross-sectional study 

Dear Dr. Hamamoto:

I'm pleased to inform you that your manuscript has been deemed suitable for publication in PLOS ONE. Congratulations! Your manuscript is now with our production department. 

Kind regards, 

on behalf of

Dr. Kanhaiya Singh 

Academic Editor

PLOS ONE